# The Role of Bioceramics for Bone Regeneration: History, Mechanisms, and Future Perspectives

**DOI:** 10.3390/biomimetics9040230

**Published:** 2024-04-12

**Authors:** Md Amit Hasan Tanvir, Md Abdul Khaleque, Ga-Hyun Kim, Whang-Yong Yoo, Young-Yul Kim

**Affiliations:** Department of Orthopedic Surgery, Daejeon St. Mary’s Hospital, College of Medicine, The Catholic University of Korea, Daejeon 34943, Republic of Korea; tanvir002@catholic.ac.kr (M.A.H.T.); abdulkhaleque.dream@gmail.com (M.A.K.); rlargus21@gmail.com (G.-H.K.); yhy30519@cmcnu.or.kr (W.-Y.Y.)

**Keywords:** bioceramics, osteoporosis, MSCs, scaffold, BMP, MAPK, Wnt/β-catenin, bone regeneration

## Abstract

Osteoporosis is a skeletal disorder marked by compromised bone integrity, predisposing individuals, particularly older adults and postmenopausal women, to fractures. The advent of bioceramics for bone regeneration has opened up auspicious pathways for addressing osteoporosis. Research indicates that bioceramics can help bones grow back by activating bone morphogenetic protein (BMP), mitogen-activated protein kinase (MAPK), and wingless/integrated (Wnt)/β-catenin pathways in the body when combined with stem cells, drugs, and other supports. Still, bioceramics have some problems, such as not being flexible enough and prone to breaking, as well as difficulties in growing stem cells and discovering suitable supports for different bone types. While there have been improvements in making bioceramics better for healing bones, it is important to keep looking for new ideas from different areas of medicine to make them even better. By conducting a thorough scrutiny of the pivotal role bioceramics play in facilitating bone regeneration, this review aspires to propel forward the rapidly burgeoning domain of scientific exploration. In the end, this appreciation will contribute to the development of novel bioceramics that enhance bone regrowth and offer patients with bone disorders alternative treatments.

## 1. Introduction

Living organisms acquire their strength, durability, and some degree of flexibility from bones, which consist of collagen and calcium phosphate apatite crystals [1]. Bone, a mineralized connective tissue, has four distinct cell types: osteoblasts, bone lining cells, osteocytes, and osteoclasts [2]. It holds pivotal roles within the body, including facilitating locomotion, providing structural support, and safeguarding delicate tissues, as well as serving as a reservoir for calcium and phosphate while also hosting bone marrow [3]. Despite that, conditions such as osteoporosis, osteoarthritis, and fractures become more common as people grow older, posing substantial challenges in clinical settings with increased demand for materials to repair bones. Efforts to address these bone-related issues have evolved from traditional autogenous and allogeneic bones to modern polymer materials and tissue-engineered bones. Ongoing scientific and clinical research in related fields is dedicated to advancing solutions for bone-related diseases [4,5]. In a clinical context, bone defects are typically treated with autologous bone grafts and allograft bone grafts. Autologous bone grafting is often regarded as the “gold standard” due to its biological benefits and cost-effectiveness [6]. Nevertheless, these methods encounter issues such as limited donor bone volume, complications after autologous bone graft procedures, and a relatively high occurrence of immune rejection associated with allograft [7,8].

Subsequently, it is crucial to have a comprehensive understanding of the intricate biomechanics of natural bone and a basic knowledge of commercially available bone grafts to facilitate successful treatment planning [9]. As a consequence, bone stands as the second most commonly transplanted tissue on a global scale, with a minimum of four million surgeries annually employing bone grafts and substitute materials [10,11,12]. Therefore, bone tissue engineering has become a promising alternative to traditional approaches to treating bone defects, offering solutions to overcome their drawbacks. Several techniques, including the utilization of bioceramics; mesenchymal stem cells; and the incorporation of diverse drugs within bioceramic substrates, employing advanced methodologies such as 3D printing and laser sintering, have been devised to fabricate composite scaffolds [13,14].

The term ‘bioceramics’ denotes biocompatible ceramic materials that find application in biomedical or clinical contexts [15]. Their chemical composition influences the degradability, biocompatibility, and bioactivity of bioceramics when implanted in vivo [16]. Various biomaterials have been employed as scaffolds, providing an option for the regeneration of bone. An ideal scaffold should exhibit an interlinked porous configuration, controlled biodegradability, mechanical characteristics akin to the bone, biocompatibility, and pores sized to facilitate cell proliferation and nutrient diffusion. The porosity of the scaffold significantly impacts cellular adherence, proliferation, revascularization, optimal nourishment, and other factors influencing cellular behavior [17,18,19]. The limitations encompass brittleness, inadequate fracture toughness, exceptionally low elasticity, and exceedingly high stiffness [19,20].

Some reports suggest that between 5% and 10% of all bone fractures, and even in some cases, up to 50%, lead to delayed or unsuccessful healing [21]. To address these challenges, several research studies have concentrated on improving bone regeneration by applying mesenchymal stromal cells (MSCs) derived from different connective tissues [22]. Methods for in vivo processing and culture have been developed to acquire an ample quantity of MSCs for therapeutic purposes [23]. Moreover, researchers have integrated MSCs with diverse scaffolds and signaling factors to engineer viable “bone substitutes” that mimic the essential characteristics of autologous bone grafts, thereby promoting enhanced bone regeneration (Figure 1) [24].

In the treatment of bone fractures and bone diseases, numerous beneficial therapeutic agents exist. However, since bones are distributed throughout the entire body, maintaining the blood concentration of therapeutic agents within a specific range is crucial to exerting their pharmacological effects at the intended peripheral site. This often leads to undesirable systemic side effects, creating a narrow toxic–therapeutic window for the treatment of bone diseases. Consequently, there is a petition for the expansion of innovative drug delivery systems to mitigate these issues [25]. In the context of treating bone fractures, sustained drug delivery systems are gaining attention. These systems consist of biocompatible and biodegradable composites, incorporating bioactive materials and ceramic implants. They are currently undergoing successful evaluation in animal models as a promising approach for sustained drug release in the treatment of bone fractures [26]. In the jurisdiction of biomaterial application, meticulous attention must be directed toward the pivotal factors governing the genesis of replacement bone tissue. These factors are bifurcated into biological determinants and mechanical stability requisites. Four fundamental components stand out in bone repair and rejuvenation: osteogenic cells, MSCs, growth factors facilitating osteoinduction, and scaffolding essential for osteoconduction.

In this review, we will explore the function of bioceramics for bone regeneration. Initially, we delve into the historical background and categorization of bioceramics. Subsequently, we examine the reactions observed in laboratory settings and the bone-forming capabilities observed in living organisms regarding the porous scaffolds’ ability to promote bone growth. Following this, we recognize how different biomolecules delivered within the scaffolds impact the integration of bone implants and the formation of new blood vessels. Lastly, we address the impact of environmental factors on bioceramics and the incorporation of drug delivery mechanisms, and we speculate on forthcoming advancements in scaffold technology for advancing bone tissue engineering practices.

## 2. History of Bioceramics

In 1920, the first documented successful use of a calcium phosphate reagent to treat a bone defect in a human patient was reported [27]. In the 1950s, the focus was on using inert materials to avoid tissue interaction, solely based on non-toxicity [28]. However, ceramics like zirconia and alumina trigger foreign body reactions, leading to encapsulation [29]. The 1960s saw significant progress in bioceramics, with notable contributions from various researchers worldwide [30]. In the 1980s, there was a shift toward using ceramics that interacted with the body, promoting bone formation [31,32,33]. Today, the aim is to create porous ceramics serving as scaffolds for tissue self-regeneration [34,35,36]. Calcium orthophosphates, especially hydroxyapatite, have been key materials since the 1980s [37]. Currently, a range of biodegradable bioceramics has been effectively applied, although with certain limitations. Consequently, further research is warranted to address these constraints.

## 3. Classification of Bioceramics

The first generation of inert ceramics aimed to substitute natural bone; hence, the research was only focused on inert materials (e.g., titanium, alumina, and polyethylene). The second one was aimed at mimicking some biomineralization-related functions, and sol–gel chemistry plays a paramount role in their synthesis and properties (e.g., hydroxyapatite or bioactive glasses). Finally, the purpose of third-generation bioceramics is basically to provide an adequate scaffolding system that helps bone cells perform their natural processes, such as biodegradable ceramics (e.g., peptide or protein-modified, degradable polymers). Tissue engineering attempts to develop artificial materials able to replace biological tissues in situations where the human body cannot perform said replacement by itself (Figure 2 and Table 1) [38].

## 4. Relationship of Different Types of Bioceramics

First-generation biomaterials include alumina and zirconia, which are exemplary for their inert bioceramic properties. These materials are still utilized in certain prosthesis components, particularly when low friction coefficients are necessary. However, these components can lead to micromovements at the bone-to-implant interface, which can escalate over time and ultimately result in prosthesis failure. In general, ceramics are inorganic materials that have a combination of ionic and covalent bonding. Bioceramics can be derived from various substances such as alumina, zirconia, magnesia, carbon, silica-containing compounds, calcium-containing compounds, and several other chemicals. Hence, compounds like calcium phosphates, calcium sulfates, specific glasses, and glass ceramics are recognized as authentic examples of bioceramics. Despite carbon being an element and exhibiting electrical conductivity in its graphite state, it is also classified as a ceramic due to its numerous ceramic-like properties. Presently, research is focused on developing new advanced bioceramics, including ordered mesoporous silica materials and specific compositions of organic–inorganic hybrids. These compounds can be manufactured in dense or porous forms, either in bulk or as crystals, powders, particles, granules, scaffolds, and/or coatings [49,50,51].

## 5. Mechanism of Action

### 5.1. The Signaling Pathway of MAPK

The signaling pathways involved in osteogenic differentiation and bone regeneration focus on MAPK pathways and the effects of biomaterials [52]. The role of extracellular stimulation in activating MAPK cascades has been highlighted, with extracellular signal-regulated kinase (ERK), c-Jun N-terminal kinase (JNK), and p38 pathways playing essential parts in inflammation, apoptosis, and growth [53,54,55]. Nano-HA accelerates osteogenic gene expression via ERK signaling, while Mg^2+^ promotes bone regeneration by activating the ERK1/2 and p38 pathways [56,57,58]. Calcium silicate (CS) releases ions that stimulate osteoblast proliferation through MAPK pathways, with silicon ions enhancing ERK and p38 activity [59,60,61,62]. Additionally, CS induces a proinflammatory response through the Toll-like receptor 2 (TLR2)-mediated JNK pathway [63]. Overall, the interaction between biomaterials and MAPK signaling pathways plays a key role in bone tissue engineering and regeneration [63].

### 5.2. The Signaling Pathway of BMP

BMPs, part of the transforming growth factor-β (T_G_F_-beta) superfamily, are essential for bone formation and stem cell differentiation [64]. They activate signaling pathways by binding to bone morphogenetic protein receptor-I (BMPR-I) and bone morphogenetic protein receptor-II (BMPR-II) receptors, leading to phosphorylation events in the small mother against decapentaplegic (SMAD) complex [65]. This activation ultimately stimulates the expression of runt-related transcription factor 2 (Runx2), a critical transcription factor controlling osteogenesis-related genes [66,67]. Bioceramics like HA and CS activate the BMP signaling pathway, promoting the osteogenic differentiation of bone marrow stem cells (BMSCs). This is evidenced by the increased expression of BMP2 and BMP4, along with key molecules in the SMAD pathway [68]. These pathways play a key role in osteoblast migration and differentiation. The inhibition of BMP2 activity leads to the downregulation of downstream cascades involving bone morphogenetic proteins [69]. On the other hand, a bioinert ceramic known as zirconium, when utilized at the nanoscale, was discovered to boost the formation of apatite when combined with polymeric scaffolds [70]. This interaction triggers BMP2 signaling by facilitating the movement of phosphorylated SMADs into the nucleus, thereby promoting osteogenesis [71].

### 5.3. The Signaling Pathway of Wingless/Integrated (Wnt)/β-Catenin

The Wnt signaling pathway is crucial for osteoblast differentiation and bone development in mesenchymal stem cells (MSCs). In the classical pathway, Wnt ligands bind to Frizzled receptors and low-density lipoprotein receptor-related protein 5 (LRP5) and low-density lipoprotein receptor-related protein 6 (LRP6), preventing β-catenin degradation and promoting osteogenic gene expression [72]. Bioceramics release ions like calcium, phosphate, and silicate to activate this pathway, aiding bone repair [73,74]. Strontium-substituted bioactive glass increases β-catenin levels in hBMSCs, while Wnt pathway inhibitors block β-catenin expression [75]. Boron-containing BG (bioactive glass) scaffolds enhance osteogenesis via transcription factor 7-like 2 and mediate the upregulation of lipocalin-2. Overall, activating the Wnt/β-catenin pathway promotes osteogenic differentiation, offering therapeutic potential in bone regeneration [76].

Furthermore, the heightened surface area facilitated by the nanocrystal trigger facilitated a greater release of calcium ions, leading to the elevated expressions of angiopoietin-1 (Ang-1) and angiopoietin-2 (Ang-2). These proteins play crucial roles in maintaining the structural integrity of blood vessels and exert antagonistic effects on each other. Ang-1 also plays a role in influencing bone formation, the production and mineralization of osteocalcin (OC), and the activity of alkaline phosphatase (ALP) (Figure 3) [77].

## 6. Uses of Porous Bioceramics

Porosity plays a key role in the fields of scaffolding and biomaterial engineering [78]. The dimensions, configurations, and arrangement of pores influence the behavior of implanted materials through biological mechanisms such as colonization, angiogenesis, vascularization, complete resorption, and replacement of implants by newly formed tissues. Therefore, cellular activities necessitate adequate space for proliferation, multiplication, the removal of toxic byproducts, and the regeneration of normal body tissues. Achieving this entails taking care to select materials that can support the desired cells, along with appropriately designed voids to facilitate penetration into biomaterials implanted in vivo [79]. However, in cases of pathological fractures or extensive bone defects, the process of bone healing and repair may be compromised. Factors such as inadequate blood supply, bone or tissue infections, and systemic diseases can adversely affect bone healing, leading to delays in unions or non-unions [80,81]. Calcium phosphates and bioactive glasses are excellent materials for constructing scaffolds intended to serve as a three-dimensional porous framework, facilitating new bone formation within their pore structure [82]. Several techniques have been employed to regulate the porosity of scaffolds (Figure 4). One effective approach involves combining freeze-drying with leaching template techniques to produce porous structures. In this method, the pore size is adjustable by controlling parameters such as the gap space of the leaching template, temperature fluctuations, and altering the density or viscosity of the polymer solution concentration during the freeze-drying process [83,84].

The dimensions of osteoblasts typically range from approximately 10 to 50 μm (µm) [85]. Indeed, osteoblasts exhibit a preference for larger pores, typically ranging from 100 to 200 μm (µm), when regenerating mineralized bone post-implantation. This size preference facilitates the infiltration of macrophages, which play a crucial role in eliminating bacteria, as well as inducing the infiltration of other cells involved in colonization, migration, and vascularization in vivo [86]. On the other hand, smaller pore sizes, typically less than 100 μm (µm), are associated with the formation of non-mineralized osteoid or fibrous tissue instead of mineralized bone [87]. Cheng et al. asserted that the utilization of magnesium scaffolds featuring two pore sizes, 250 and 400 µm, revealed that the larger pore size fosters the enhanced formation of mature bone by facilitating vascularization. This is attributed to newly formed blood vessels, which deliver ample oxygen and nutrients necessary for osteoblastic activity within the larger pores of the implanted scaffolds. Thus, this upregulates the expression of osteopontin (OPN) and collagen type I, leading to the subsequent generation of bone mass [88]. Shrestha et al. demonstrated that a phosphate-enriched nanocomposite comprising titanium and zinc exhibited remarkable potential for bone regeneration in vitro. This composite induces the robust activation of MC3T3-E1 and hBM-MSCs cells, thereby instigating a profound enhancement in cell viability and fostering osteogenic differentiation. This is substantiated by the notable upsurge in the expression levels of critical bone-related markers such as ALP, Col1a1, RUNX2, OPN/Spp1, and OCN. Moreover, in vivo assessments further accentuate the significant augmentation of fresh bone formation observed within critical-size calvarial defects in a rat model, thus underscoring the unprecedented efficacy of this bioceramic composite in facilitating bone regeneration [89]. In another study, Cao et al. created porous composite scaffolds, comprising polyglycolic acid/beta-tricalcium phosphate (PGA/β-TCP) in weight ratios of 1:1 and 1:3, using a sophisticated blend of solvent-forming and particulate leaching methods. After their insertion into rats, these scaffolds underwent rigorous evaluation through quantitative imaging analysis and qualitative histological assessments. Their results revealed that bone regeneration commenced within 14 days post-surgery and progressed exceptionally well, with significant healing apparent by the 30-day mark. By the 90-day milestone, bone replacement was nearing completion, showcasing a healthy bone appearance. Remarkably, the scaffold with a 1:3 ratio of polyglycolic acid (PGA) to beta-tricalcium phosphate (β-TCP) demonstrated remarkable osteogenic, mineralization, and biodegradation properties [90]. Huang et al. proclaimed that porous poly(lactide-co-glycolide/nanohydroxyapatite (PLGA/NHA) composite scaffolds use the thermally induced phase separation technique. They examined the impacts of solvent composition, polymer concentration, coarsening temperature, coarsening time, and nano-HA content on the micromorphology and mechanical properties of the scaffolds. Their findings demonstrated that the inclusion of nano-HA significantly enhanced the mechanical properties and water absorption capacity of the scaffolds. Additionally, the PLGA/nano-HA scaffolds exhibited substantially greater cell growth and ALP activity compared to scaffolds without nano-HA [91]. In a study by Lee et al., at the 6-month follow-up, a significant fusion rate and positive clinical outcomes were observed in the reconstruction of malar defects using patient-specific 3D-printed BGS-7 implants. Additionally, all participants in the trial expressed satisfaction with both the aesthetic and functional outcomes following the operation with BGS-7 implants. Consequently, this study underscores the safety and potential efficacy of utilizing patient-specific 3D-printed BGS-7 implants for facial bone reconstruction, suggesting promising value in this approach [92].

Torres et al. reported that three types of scaffolds were made from a combination of β-TCP and HA nanopowder at different ratios (80/20%, 90/10%, and 99/1%). An alginate coating was applied to enhance mechanical strength and mimic the native bone matrix. The 80/20 scaffold had the highest porosity and mechanical properties. Coating improved mechanical strength, with the 80/20/A scaffold showing the best performance. All scaffolds demonstrated high biocompatibility with human osteoblast cells, with the 80/20 formulation being the most compatible. SEM analysis confirmed cell migration within the coated scaffolds. Overall, the 80/20 scaffold, especially when coated with alginate, showed promising properties for bone tissue engineering [93]. Fukuda et al. implanted cylinders with a scaffold comprising longitudinal square channels of varying diagonal widths (500, 600, 900, and 1200 μm) into the dorsal muscles of beagle dogs. These implants were sterilized using ethylene oxide gas and observed over 16, 26, or 52 weeks. Their results showed excellent bone formation along the channels, with more bone quantity in implants with smaller diagonal widths (p500 and p600). Significant osteoinduction was observed in p500 and p600, with the greatest effect seen at 5 mm from both ends of the implants. Thicker laminated bone formation was noted at later time points (26 and 52 weeks), indicating a time-dependent increase in bone formation [94]. Nevertheless, second-generation biomaterials present promising forecasts for scaffold fabrication. Specifically, bioceramic scaffolds must adhere to specific porosity standards, aligning closely with those found in natural bone tissue [28].

## 7. Application of Adipose-Derived MSCs

Adipose-derived mesenchymal stem cells (ADMSCs) possess an undifferentiated immunophenotype and exhibit characteristics such as self-renewal and multipotency. This means they express specific surface markers typical of MSCs and can undergo numerous cell divisions while retaining their undifferentiated state, with the ability to differentiate into various mesodermal cell types like adipocytes, osteoblasts, and chondrocytes. These properties make ADMSCs promising candidates for applications in regenerative medicine and tissue engineering owing to their abundant presence in adipose tissue, ease of isolation, and potent regenerative capabilities [95,96,97,98,99,100,101]. Furthermore, ADMSCs can release numerous anti-inflammatory proteins in response to inflammatory stimuli, including tumor necrosis factor-α (TNF-α), interleukin-4 (IL-4), IL-6, IL-10, and IL-1 receptor antagonists. Additionally, these cells can produce a diverse array of growth factors, such as T_G_F_-β1, hepatocyte growth factor (HGF), vascular endothelial growth factor (V_E_G_F_), and stromal cell-derived factor (S_D_F_-1). These factors play pivotal roles in tissue remodeling, angiogenesis, and antiapoptotic processes [100,101,102].

Research on ADMSCs in the field of tissue engineering has shown significant promise. In recent studies by Rivera-Izquierdo et al., it has been demonstrated that ADMSCs hold potential applications in addressing various musculoskeletal disorders and cartilage-related conditions. These include, but are not limited to, osteoporosis, osteonecrosis, fractures, osteoarthritis, and cartilage lesions [103]. McCullen et al.’s results suggest that adding 10% tricalcium phosphate (TCP) to electrospun poly(L-lactic acid) (PLA) enhances the proliferation of human adipose-derived stem cells (hASCs). Conversely, incorporating 20% TCP into electrospun PLA promotes osteogenic differentiation and boosts calcium accumulation by hASCs when compared to pure electrospun PLA scaffolds [104]. School et al. demonstrated that, after 8 weeks, histological examination revealed substantial tissue regeneration in hypothyroid rats treated with ADMSC-conditioned media (ADMSC-CM) compared to untreated rats. Specifically, the newly formed tissue almost completely covered the defect caused by hypothyroidism in the group treated with ADMSC-CM. Their findings suggest that combining ADMSC-CM with bioceramic collagen could effectively promote bone repair in hypothyroid patients with compromised bone regeneration capabilities [105]. Xia et al. demonstrated that highly interconnected macroporous HAp scaffolds, characterized by diverse surface topographies such as nanosheets, nanorods, and micro–nanohybrids, effectively promoted the attachment, spreading, proliferation, and osteogenic differentiation of rat adipose-derived stem cells (ASCs). Additionally, these scaffolds were found to upregulate the expression of angiogenic factors, indicating their potential to enhance angiogenesis alongside osteogenesis. Subsequently, in vivo bone regeneration assessments using rat critical-size calvarial defect models validated that the combination of the micro–nanohybrid surface and ASCs significantly augmented both osteogenesis and angiogenesis compared to the control HAp bioceramic scaffold possessing a traditional smooth surface [106]. Daei-Farshbaf et al. investigated the efficacy of combining Bio-Oss and collagen type I gel as a scaffold with human adipose tissue-derived mesenchymal stem cells (AT-MSCs) for bone regeneration in rat critical-size defects. After 8 weeks, no adverse effects were observed, and imaging analysis revealed enhanced bone regeneration in rats treated with Bio-Oss–gel compared to untreated rats. Additionally, MSC-seeded Bio-Oss–gel showed the highest level of bone reconstruction, with histological staining confirming impressive osseointegration. Overall, the combination of AT-MSCs, Bio-Oss, and gel demonstrated synergistic effects in promoting bone regeneration, suggesting potential applications in bone regenerative medicine and tissue engineering [107]. Sandor et al. mentioned that in 13 consecutive cases of craniomaxillofacial hard-tissue defects across various anatomical sites, autologous adipose tissue was used to harvest ASCs. These ASCs were cultured, expanded, and seeded onto resorbable scaffold materials for implantation into the defects. The scaffolds were either bioactive glass or β-TCP, with some cases supplemented by recombinant human bone morphogenetic protein-2, and follow-up periods ranged from 12 to 52 months. Ultimately, ten patients were successfully treated [108].

In a clinical study, Prins et al. investigated the efficacy of stromal vascular fraction (SVF) without in vitro cell culture. Ten patients requiring maxillary sinus floor elevation received treatment with freshly isolated SVF extracted from autograft fat tissue using a Celution 800/CRS device. The SVF was transplanted with ceramics to increase vertical bone height in the posterior maxilla. Panoramic radiographs showed no significant difference between the control and study sides. However, microcomputed tomography and biopsy evaluations revealed significantly greater osteogenesis on the stem cell-transplanted side [109].

In summary, ASCs offer the advantages of easy access and abundant supply [110]. While ASCs offer benefits such as easy access and ample supply, ex vivo expansion is typically necessary to minimize contamination with other cell types. However, it is important to note that ASCs cultivated in vitro have demonstrated reduced stemness, self-renewal, or multipotency compared to their native state [111]. Furthermore, the safety of ASCs has not been definitively established. Chromosomal abnormalities have been detected in cultured ASCs, which raises concerns regarding their safety for therapeutic use [112]. When compared to bone marrow-derived mesenchymal stem cells (BMSCs), ASCs have exhibited inferior osteogenicity in laboratory settings (in vitro). However, their superiority in vivo (in living organisms) is still uncertain and requires further investigation [113]. Despite this, the clinical strategy for regeneration incorporating cell therapy and tissue engineering relies on the utilization of third-generation materials [28].

### 7.1. Application of Bone Marrow-Derived MSCs

BM-MSCs have been thoroughly examined and understood regarding their phenotype and biological characteristics. They have attracted considerable attention due to their abundance in bone marrow and their well-documented capacity to differentiate into multiple cell types, including osteoblasts, adipocytes, and chondrocytes. Scientists have extensively studied the cell surface markers and immunomodulatory properties of BM-MSCs, as well as their secretion of trophic factors. These findings have contributed to understanding their potential therapeutic applications in various diseases and tissue regeneration approaches. Overall, the detailed investigation of BM-MSCs has provided a strong basis for leveraging their therapeutic potential in regenerative medicine and tissue engineering [114,115].

Neen et al. carried out a prospective case–control study, in which the efficacy of a collagen/hydroxyapatite matrix infused with BMSCs was compared to that of conventional iliac autografts. The BMSC-infused matrix demonstrated similar effectiveness in promoting posterolateral fusion but exhibited comparatively lower efficacy in facilitating underbody and 360° fusions. Despite the potential drawbacks in fusion outcomes, the use of BM-MSCs offered a significant advantage by reducing the risk of donor site complications such as pain and neuroma formation [116]. In a study by Quarto et al., the initial documentation of utilizing culture-expanded BM-MSCs in conjunction with hydroxyapatite biomaterial for addressing substantial bone defects arising from traumatic fractures and unsuccessful lengthening was reported. In this study, three patients were treated, and notable outcomes included significant callus formation surrounding the implants and favorable integration observed at the interfaces with the host bones within two months post-surgery. Furthermore, there were no reported adverse reactions to the implants, and all three patients successfully regained full limb functionality [117]. Numerous registered clinical trials have employed autologous bone marrow mononuclear cells (BM-MNCs), MSCs, or preosteoblasts, either administered via injection or coapplied with bone auxiliary materials as transporters, to address delayed unions and non-unions of extended bones. Among these studies, Emadedin et al. documented the safety of injecting cultured MSCs, with evidence of bone union observed in three out of the five treated patients [118]. Moreover, Gomez-Barrena et al. documented that the surgical administration of culture-expanded bone marrow MSCs in conjunction with bioceramic granules for addressing delayed unions and non-unions was both safe and viable. Their study revealed that 26 out of 28 patients showed radiological evidence of healing one year after treatment [119].

Xu et al. performed a study in which, initially, MSCs were cultured for two weeks before being introduced into the damaged area. Subsequently, these cells were placed onto a bioactive glass–collagen–hyaluronic acid phosphatidylserine scaffold. The presence of MSCs notably boosted the generation of new bone tissue [120]. Zhang et al. reported that, following a 7-day culture period, (Sr_2_ZnSi_2_O_7_) SZS and (Sr_2_MgSi_2_O_7_) SMS significantly enhanced the osteogenic differentiation of BMSCs compared to conventional β-TCP. Furthermore, in terms of their overall effectiveness, SZS and SMS exhibited comparable abilities to akermanite (Ca_2_MgSi_2_O_7_) and CMS in promoting cell growth and stimulating cell differentiation. Their results collectively suggested that SZS and SMS hold promise for use in bone regeneration applications [121].

Sun et al. demonstrated that, in addition to the enhancement of cell proliferation, akermanite promoted the osteoblastic differentiation of hBMSCs in vitro by upregulating osteogenic gene expression. Akermanite bioactive ceramics not only simply boosted the osteoblastic differentiation of hBMSCs in osteogenic media (a-MEM supplemented with ascorbic acid, glycerophosphate, and dexamethasone) but also improved cell proliferation in normal growth medium without osteogenic reagents. These results suggest that akermanite may be used as a more promising bioactive ceramic for bone regeneration and tissue engineering applications [122]. Xia et al. reported that akermanite bioceramics have several benefits for bone regeneration in osteoporosis. They enhance cell growth and differentiation, increase angiogenic factor expression, and suppress osteoclast activity. In animal models, they outperform β-TCP bioceramics in promoting bone formation, angiogenesis, and inhibiting osteoclastogenesis. The presence of magnesium and silicon ions in akermanite bioceramics contributes to these positive effects, making them promising for osteoporotic bone regeneration [123]. Maiti et al. performed a study on live rabbits using a critical-size defect (CSD) model to confirm the application of BMSCs whether obtained from the same individual (autogenous), a genetically distinct individual of the same species (allogenous), or even from a different species (xenogenous) and elaborated that seeding onto bioscaffolds accelerates the healing process of critical-size defects. Therefore, it can be inferred that BMSCs show potential for promoting bone formation in situations like fracture healing and non-unions [124]. Lin et al.’s studies involved creating macroporous scaffolds using strontium-containing calcium silicate (SrCS) bioceramics. Their findings showed that SrCS materials, which release bioactive strontium and silicon ions, created an environment conducive to directing rBMSCs-OVX toward becoming osteoblasts and promoted angiogenesis in endothelial cells (ECs). Both pure CS and SrCS inhibited osteoclastogenesis by stimulating osteoprotegerin (OPG) and suppressing the receptor activator of nuclear factor kappa-Β ligand (RANKL). However, the inhibitory effect was stronger and longer-lasting with SrCS compared to CS, which only showed early-stage inhibition [125]. Maiti et al. demonstrated that cultivating autologous rabbit bone marrow-derived mesenchymal stem cells (rBMSCs) on a bioscaffold made of silica-coated calcium hydroxyapatite (HASi) could expedite the scaffold’s osteoconductive properties, presenting a potential substitute for autogenous bone grafts in addressing substantial bone defects and non-healing fractures. Additionally, introducing growth proteins, specifically recombinant human bone morphogenetic protein (rhBMP-2), to the autogenous MSC-seeded HASi bioceramic framework could accelerate bone regeneration in a rigorously controlled study using a large critical-size bone defect model [126].

In the study by Gomez-Barrena et al., effective bone consolidation utilizing expanded hBM-MSCs in combination with biomaterials, as assessed through clinical and radiological examinations, was particularly emphasized during a 12-month assessment. Bone regeneration was additionally confirmed via bone biopsies. There were no notable disparities observed in the consolidation of affected bones, although tibial non-unions displayed slower consolidation rates. Individuals who smoked exhibited reduced consolidation scores at both the 6- and 12-month intervals, while no noticeable impact was identified based on gender or the duration since the initial fracture [127].

### 7.2. Application of Extracellular Vesicle-Derived MSCs

Intercellular communication, crucial for multicellular organisms, traditionally occurs through direct cell–cell contact or the transfer of secreted molecules. However, a third mode of communication has emerged in the last two decades, involving the exchange of EVs between cells. These EVs, encompassing exosomes and microvesicles, have gained prominence for their roles in liquid biopsy as biomarkers and their potential therapeutic applications. EVs, by ferrying a diverse array of biomolecules, can influence the behavior and function of recipient cells, impacting various physiological and pathological processes. Their presence in bodily fluids offers opportunities for non-invasive disease diagnosis and monitoring. Moreover, EV-based therapies hold promise in regenerative medicine, drug delivery, and immunotherapy. In essence, the discovery and understanding of EVs have broadened our insights into intercellular communication and opened new avenues for medical diagnostics and treatments [128]. Particularly, the therapeutic application of EVs is involved in bone regeneration, thanks to the regulation of immune environments, the enhancement of angiogenesis, the differentiation of osteoblasts and osteoclasts, and the promotion of bone mineralization. Since naked EVs are vulnerable when transplanted in vivo and difficult to target at bone defect sites, the approach of loading EVs with biomaterial systems possesses tremendous advantages, as shown in the schematic illustration in Figure 5.

Indeed, EVs delivered via biomaterials show great promise in the field of bone regeneration. These EVs can be immobilized within gels, actively linked to molecular binders, or affixed to the surfaces of biomaterials, enabling precise control over their release. This controlled release mechanism holds the potential for enhancing bone healing and regeneration processes [129]. Zhang et al. mentioned that the integration of exosome/β-TCP combination systems enhanced osteogenesis ability compared to the utilization of β-TCP scaffolds alone in a rat critical-size calvarial bone defect model [130]. Wiklander et al. also illustrated that they attained alveolar bone regeneration using an exosome/β-TCP system [131].

As reported in a recent review and research paper, EVs derived from osteoblasts, osteoclasts, osteocytes, monocytes, macrophages, and dendritic cells contain various miRNAs and proteins. These components are implicated in either enhancing or inhibiting osteogenic activity [132]. Research has indicated that extracellular vesicle (EV)-borne miRNA can be targeted by high-mobility group AT-hook 2 (HMGA2) [133], glycogen synthase kinase-3β/β-catenin [134], or the Wnt/β-catenin pathway [135], all of which are known to contribute to osteogenic differentiation. Furthermore, EVs also carry abundant proteins such as tissue non-specific alkaline phosphatase (TNAP6) and matrix metalloproteinase (MMPs), which are essential for bone remodeling.

## 8. Impact of Environmental Factors on Bioceramics

In recent eras, a superfluity of groundbreaking biomaterials with the capacity for remote activation has emerged, presenting an inimitable prospect to precisely and locally treat diseases by modulating cell signaling pathways in vitro [136,137,138,139]. Ophthalmic, electrical, ultrasound, and magnetic methodologies have been widely deployed to encourage synthetic biomaterials to modulate cell signaling, leveraging their non-invasive attributes and precise aptitude to regulate biological functions. These modalities reveal significant potential for translation into in vivo surroundings [140,141,142,143,144]. Amid the assortment of approaches, near-infrared (NIR) light stimulation stands out for its expedient features, as it imposes minimal detriment upon cells and organs.

Fu et al. elaborated that utilizing near-infrared (NIR) light to activate photoelectrons within a bismuth sulfide/hydroxyapatite (BS/HAp) film effectively guides cellular fate toward osteogenic differentiation in vitro and promotes improved bone regeneration in vivo. The engineered Ti-BS/HAp composite demonstrated elevated photocurrent levels, credited to the reduction in h+ ions and the transfer of interfacial charges facilitated by HAp. This mechanism empowers the regulation of biological processes within deep tissues using NIR light [145].

The physical and chemical features of bioceramics are subject to amendment in environments categorized by varying pH levels [146]. The instigation of hydration, the formation of hydrate phases, and the reactions of cement clinkers can be obstructed by the concentration of hydrogen ions and the pH level [147]. Acute mechanical and chemical features of a biomaterial, such as setting reaction, compressive and tensile strength, hardness, and microleakage, could be conceded in acidic or alkaline environments [148,149,150]. It appears that the pace of pH fluctuations during the initial stages of hydration might influence the setting procedure and crystallization of bioceramics [151].

## 9. Drug Delivery

Poly(methyl methacrylate (PMMA), the primary material used for local antibiotic carriers in clinical settings, has several disadvantages. Firstly, heat generated during scaffold synthesis can lead to the degradation of the antibiotic and surrounding tissues. Secondly, PMMA is non-absorbable, requiring surgical removal, which increases the risk of infection. Additionally, PMMA may hinder bone regeneration and conduction, adding complexity to clinical outcomes [152,153]. Hence, there is a growing interest in utilizing absorbable and osteogenic novel materials for fabricating antibiotic-loaded carriers [154].

For that reason, the incorporation of antibiotics into osteoconductive materials such as calcium sulfate, hydroxyapatite, and tricalcium phosphate for localized osteomyelitis treatment has shown promise, effectively addressing the issue of dead space while simultaneously eradicating the infection [155,156,157,158,159,160]. Ongoing research aims to develop bioceramics capable of releasing antibiotics over a sufficient duration to effectively treat the infection. However, these bioceramics are designed to cease drug delivery at a specific time point to prevent low antibiotic concentrations and the development of bacterial resistance. In in vivo experiments using animal models with infected proximal tibial defects, the implantation of the vancomycin-loaded bone-like hydroxyapatite/poly(amino acid) (V-BHA/PAA) scaffold after surgical debridement revealed several positive outcomes. Not only was the infection effectively cleared, but the V-BHA/PAA scaffold was also closely integrated with the host bone. Additionally, there was an observed ingrowth of new vessels and trabecular bone around the scaffold. As the scaffold degraded over time, new bone formation gradually replaced it, demonstrating excellent osteoconductive properties. Moreover, the absence of necrosis in the surrounding bone indicated good osteogenic properties of the scaffold, with simultaneous new bone formation occurring at the defect site [161]. The vancomycin released from the V-BHA/PAA scaffold, a novel vancomycin-loaded bone repair material, demonstrated potent antibacterial activity both in vitro and in vivo against both regular S. aureus and MRSA (methicillin-resistant Staphylococcus aureus) strains. Furthermore, the antibacterial efficacy of the V-BHA/PAA scaffold surpassed that of V-PMMA significantly. Given its biodegradability and ability to induce osteogenesis, V-BHA/PAA stands out as a promising option for the treatment of chronic bone infections [162]. HAC is a bone cement used in vivo that fully converts to hydroxyapatite and exhibits excellent biocompatibility. By incorporating antibiotic solutions, such as gentamicin, HAC can serve as a carrier for various antibiotics. Mixing gentamicin into HAC does not compromise the effectiveness of the antibiotic or the mechanical properties of the cement within the concentration range typically used. This delivery system provides the sustained release of high levels of antimicrobial agents, resulting in long-term local antibacterial efficacy. Our in vivo study demonstrated a beneficial impact on bone infection resolution and regeneration [163]. In a rat model of debrided osteomyelitis, ceramic void filler containing gentamicin (CERAMENT G) demonstrated a decreased rate of persistent infection and enhanced new bone growth compared to the same void filler lacking antibiotics (CERAMENT) and an empty defect. Additionally, tobramycin-incorporated calcium phosphate bone substitute (CPB) effectively facilitated soft tissue and bone healing while preventing early bacterial colonization by *Staphylococcus aureus* in infected tibia [164].

## 10. Limitations and Future Potential Development of Bioceramics

The primary drawback of bioceramics lies in their suboptimal mechanical properties, particularly their limited reliability when subjected to tensile loads. Bioceramics exhibit low toughness, inadequate fatigue resistance, brittleness, and impact resistance, which significantly restricts their suitability for serving as bulk materials capable of withstanding tension or impact forces [165,166,167]. Incorporating bioceramics into composite materials with other reinforcements enhances mechanical properties. Techniques such as nanoparticle doping and nanofiber reinforcement improve strength, toughness, and ductility at the nanoscale. In addition, introducing dopant elements such as magnesium or strontium enhances mechanical properties and bioactivity. The sintering process control during fabrication influences microstructure and mechanical properties. The optimization of parameters, including temperature and time, enhances density, strength, and fatigue resistance.

The drawback of scaffold research lies in the absence of specific recommendations regarding which type of scaffold—whether macro, meso, or micro—is best suited for different types of bones. Nevertheless, the choice of scaffold for bone fracture healing depends on factors such as bone position, mechanical properties, and biological requirements, as well as scaffold material features. Long bone fractures may require stable macroporous scaffolds, while flat bones could benefit from microporous scaffolds with controlled pore sizes. Irregular bones might need mesoporous scaffolds accommodating different cell types. Customized scaffold designs tailored to specific pore structures and material properties may be necessary for optimal healing. Researchers should carefully consider these factors and conduct comprehensive studies to assess scaffold effectiveness across different bone types and anatomical locations.

Additionally, a promising avenue for advancing bioceramics for bone regeneration involves combining bioactive ceramics with drug delivery and biodegradable materials, along with the use of extracellular vesicles. This innovative approach holds significant potential for developing future generations of effective bioceramics for bone healing.

## 11. Conclusions

Our review thoroughly scanned the role of bioceramics in bone regeneration, covering their history, classifications, and how they work. We explored how bioceramics combined with scaffolds, cells, and drugs can aid regeneration, especially when used in porous forms. Despite progress, research in this area still leaves so many gaps and many questions unanswered. So it is crucial to keep investigating to unlock bioceramics’ full potential. Recent studies have shown promise to utilize porous bioceramic scaffolds for bone regeneration, but we need to determine which type of porous scaffold is best fit for different parts of the body. Although bioceramics hold great promise for bone tissue engineering, they still have weaknesses like being too brittle. Techniques like nanoparticle doping and nanofiber reinforcement offer hope for making bioceramics stronger for clinical use. Ongoing research is crucial for improving bioceramics and making them better at helping bone healing. In conclusion, gaining a more profound comprehension of the involvement of bioceramics would significantly enhance the development of more potent and efficacious therapeutic approaches for bone regeneration.

## Figures and Tables

**Figure 1 biomimetics-09-00230-f001:**
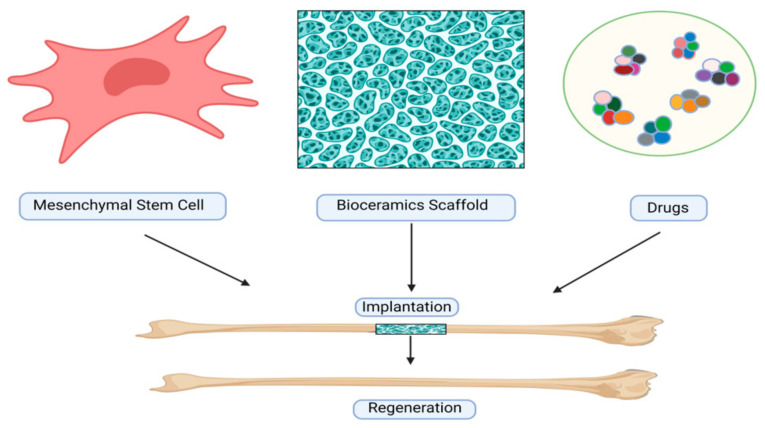
A schematic representation of the action of bioceramic scaffolds, mesenchymal stem cells, and drug delivery for bone regeneration.

**Figure 2 biomimetics-09-00230-f002:**
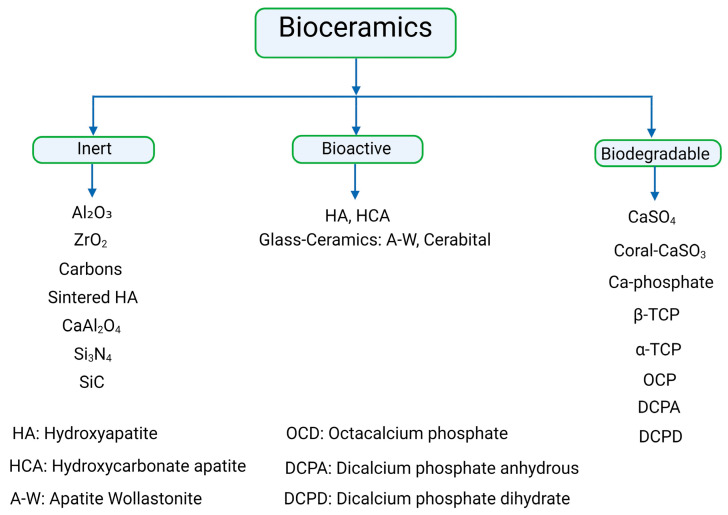
The three generations of bioceramics are laid out.

**Figure 3 biomimetics-09-00230-f003:**
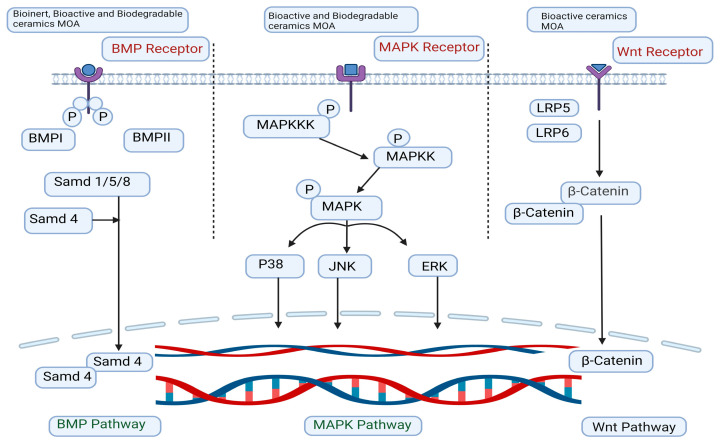
Diagram illustrating the traditional bioceramics’ cellular route.

**Figure 4 biomimetics-09-00230-f004:**
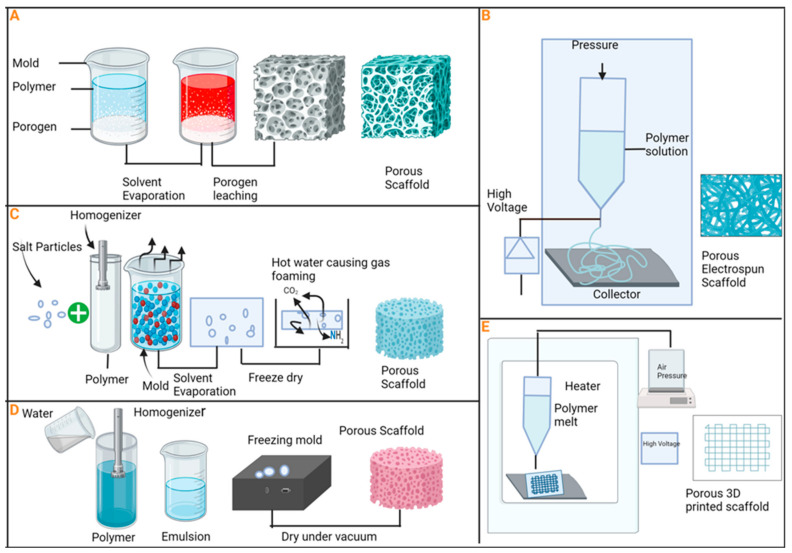
There are different approaches to manufacturing porous scaffolds: (**A**) porogen leaching; (**B**) solution electrospinning; (**C**) gas foaming; (**D**) freeze-drying; (**E**) melt electrowetting and 3D printing.

**Figure 5 biomimetics-09-00230-f005:**
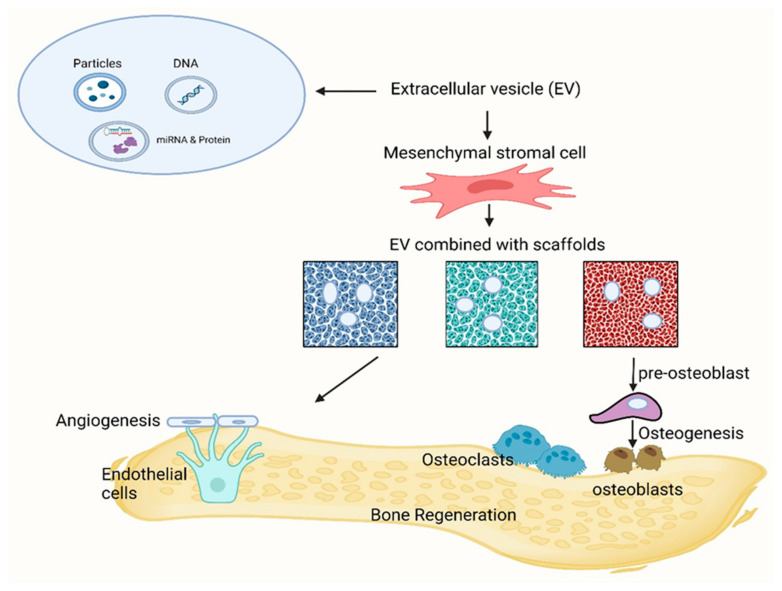
An illustrated synopsis of EVs’ role in bone regeneration.

**Table 1 biomimetics-09-00230-t001:** Different types of bioceramics and their applications with biological behavior.

Name of the Materials	Applications	Biological Behavior	References
Alumina	Femoral balls, inserts of acetabular cups, artificial heart valves, dental roots, bone screws, and endoscope	Bioinert	[39]
Al_2_O_3_	Coatings for tissue growth: orthopedic	Bioinert	[39]
Zirconia (Y-TZP)	Femoral balls, dental veneers, and tooth inlays	Bioinert	[39]
Titanium nitride	Antiwear coating of femoral balls and knee prostheses	Bioinert	[40]
Zirconium nitride	Antiwear coating of femoral balls and knee prostheses, and coating for coronary stents	Bioinert	[40]
Silicon nitride	Antiwear coatings of femoral balls	Bioinert	[41,42]
Hydroxyapatite	Bone cavity fillings, ear implants, vertebrae replacement, hip implant coatings, bone scaffolds, and orthopedic	Bioactive	[43,44]
Bioglass	Bone replacement	Bioactive	[45]
Tricalcium phosphate	Bone replacement	Bioactive/biodegradable	[46]
Hydroxyapatite/PCL	Tissue engineering scaffolds	Biodegradable	[46]
β-Tricalcium phosphate (β-TCP)	Bone regeneration, bioactivity	Biodegradable	[47]
Dicalcium phosphate dehydrate (DCPD)	Bone regeneration, osteoconductivity	Biodegradable	[47]
Calcium phosphate	Promotes tissue ingrowth and vascularization	Biodegradable	[48]

## Data Availability

The authors confirm that the data supporting the findings of this study are available within the article.

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
