# Peer review of "The Role of Bioceramics for Bone Regeneration: History, Mechanisms, and Future Perspectives"

_biomimetics, 2024, doi:10.3390/biomimetics9040230_

Round 1

Reviewer 1 Report

Comments and Suggestions for Authors

The article has been written well and i would recommend for publication 

Author Response

Thank you, and we greatly value your kind words. I express my gratitude for your glowing review.

Reviewer 2 Report

Comments and Suggestions for Authors

Comments:

This review aims to add to the expanding field of scientific research by offering a detailed look at how bioceramics and growth factors contribute to bone regeneration. This research work is very interesting, and the reviewer has some questions and suggestions that the author should take on board. 1. The introduction section does not introduce the research status of growth factor. 2. It is inappropriate for “history of bioceramics” include “mechanism of action”, “classification of bioceramics” and “application of adipose-derived mscs”, which make the paper chaotic. 3. The section “bioceramics containing growth factor seeding” does not indicate association between bioceramics and growth factors. 4. Add discussion section: structure of bioceramics and the influence of environment on bioceramics. 5. Each chapter should discuss the internal relationship between the other, which can enhance the logic of the paper. 6. Figure 2. (the three generations of bioceramics layout) should be summarized as a table containing the properties and parameters of the various materials. 7. As a review, it should have a certain number of graphics and text. Obviously, the article needs further improvement by increasing the content of images and the number of texts.

8. The paper may be helpful for enriching the content of the manuscript. (DOI: 10.1016/j.addma.2022.102694, DOI: 10.1021/acsnano.9b08115)

9. English is very poor and recommended to seek help from experts to modify English.

Comments on the Quality of English Language

English is very poor and recommended to seek help from experts to modify English.

Author Response

We greatly appreciate your insightful feedback, and we are committed to implementing as many of your suggestions as feasible. Any aspects that are presently beyond our reach will be prioritized for integration into our forthcoming projects.

Reviewer 3 Report

Comments and Suggestions for Authors

Generally, the science is reasonable. Following suggestions can improve the manuscript:

1. In the introduction, last sentence of the first paragraph ends as ‘allografts gratfts’; please change it to ‘allografts’.

2. Line #50: sentence structure needs to be changed. The word ‘method’ cannot be used to detail materials and cells.

3. Line # 60, 61, 62: sentence structure needs revision.

4. Section 2.1 starts with the word ‘besides’, that can be taken off.

5. Abbreviations: throughout the manuscript, the abbreviations have been re-defined again and again. Authors need to once define an abbreviation and continue using it afterwards. Additionally, for growth factors, the abbreviations’ definition is in the brackets and abbreviations themselves are out of the paratheses after the brackets such as this: (transforming growth factor-β) T_G_F_-beta. Either the abbreviations go before the brackets, such as T_G_F_-beta (transforming growth factor-β) or the brackets go around the abbreviations the first time they appear, such as transforming growth factor-β (T_G_F_-beta).

6. Section 2.2.4: starts with discussing the porosity of bioceramics and eventually deviates to the effect of ratios of polymers on osteogenesis. The section needs to be revised to contain only the effects of porosity on osteogenesis and relate it to bioceramics porosity. A new section can be created for polymer ratios.

Comments on the Quality of English Language

English writing needs improvement through out the manuscript. Some points have been highlighted above.

Author Response

We extend our gratitude for your valuable feedback. We endeavor to accommodate all of your suggestions to the fullest extent possible.

Reviewer 4 Report

Comments and Suggestions for Authors

This review article entitled "Role of Bioceramics and Growth Factor for Bone Regeneration" details current trends in the field of biomaterials more specifically bioceramics intended for bone regeneration. The review summarises the important bioceramics and the growth factors involved for bone tissue regeneration applications.

The review is well written and the flow of the review is readable for the readers including schematic and figures illustrated. The only minor concern in the review is to include a few more recent references (past three years) and text format of the references. I would suggest the authors to use tools like mendely or endnote to use the style prescribed by the MDPI for references.

Author Response

Thank you for your feedback. We have incorporated references from recent years as per your recommendation.

Round 2

Reviewer 2 Report

Comments and Suggestions for Authors

Continue to revise abstracts and English

Comments on the Quality of English Language

Modify English grammar and spelling errors, such as spaces between units and Arabic numerals.

Author Response

Thank you for your valuable comments. I acknowledge your consideration and have made every possible change to the manuscript.
